# Determination of Moisture Content and Shrinkage Strain during Wood Water Loss with Electrochemical Method

**DOI:** 10.3390/polym14040778

**Published:** 2022-02-16

**Authors:** Zongying Fu, Hui Wang, Jingpeng Li, Yun Lu

**Affiliations:** 1Key Laboratory of Wood Science and Technology of National Forestry and Grassland Administration, Research Institute of Wood Industry, Chinese Academy of Forestry, Beijing 100091, China; zyfu@caf.ac.cn; 2State Key Laboratory of Animal Nutrition, Institute of Animal Science, Chinese Academy of Agricultural Sciences, Beijing 100193, China; wanghui_lunwen@163.com; 3Key Laboratory of High Efficient Processing of Bamboo of Zhejiang Province, China National Bamboo Research Center, Hangzhou 310012, China; lijp@caf.ac.cn

**Keywords:** electrochemical method, moisture content, shrinkage strain, wood water loss

## Abstract

Moisture content and shrinkage strain are essential parameters during the wood drying process. The accurate detection of these parameters has very important significance for controlling the drying process and minimizing drying defects. The presented study describes an electrochemical method to determine wood moisture content and shrinkage strain during drying, and the accuracy of this method is also evaluated. According to the results, the electrical resistance of the samples increased with the decrease in wood moisture content. As the moisture content changed from 42% to 12%, the resistance increased from 1.0 × 10^7^ Ω to 1.2 × 10^8^ Ω. A polynomial fitting curve was fitted with a determination coefficient of 0.937 to describe the relationship between moisture content and electrical resistance. In addition, both the shrinkage strain and resistance change rate increased with the decrease in wood moisture content, especially for the moisture content range of 23% to 8%, where the shrinkage strain and resistance change rate increased by 4% and 30%, respectively. The shrinkage strain increased exponentially with the increase in the resistance change rate; thereby, an exponential regression equation was proposed with a determination coefficient of 0.985, expressing the correlation between the two. This demonstrates the feasibility of the electrochemical method for measuring wood moisture content and shrinkage strain.

## 1. Introduction

Wood drying is an essential step in wood processing, and it is also the most energy- and time-demanding step. When the drying quality of wood is ensured, the reduction in time and energy consumption can result in economic benefits. This enhancement requires persistent improvements of the process to obtain the best trade-off between the drying rate and quality, which can also be facilitated by a greater understanding of the drying process, especially for moisture content (MC) and shrinkage strain.

Wood MC is one of the most important parameters that need to be rapidly and accurately measured during the drying process, because both the control and adjustment of drying environmental conditions are mainly dependent on the MC. The oven drying method is one of the most traditional testing methods of wood MC, which has high precision, but the speed of the testing is slow, and continuous online testing is unachievable. The probe test method is the most popular approach in the wood processing enterprise due to the advantages of quick measurement and data evaluation. However, the method has certain limitations, as the testing accuracy is mainly concentrated on an MC below 30% and is also influenced by the temperature, probe depth and probe location, among other factors. As the measurement methods become increasingly advanced, some non-destructive methods, such as the X-ray densitometry method [1,2], X-ray microscopy [3,4], computed tomography scan [5,6] and the nuclear magnetic resonance approach [7,8], are applied to measure wood moisture content. Although the above advanced measuring methods have the characteristics of a fast measurement speed, high precision and imaging capabilities, some limitations still exist such as their high cost, their complex operation and a demanding application environment in industrial production.

Shrinkage strain is another characteristic of significant interest that can be used as an indicator to evaluate the wood drying process. It can be determined by the traditional slicing and strain gauge methods. For the slicing method, a manual measurement error is inevitable because of the contact measurement by a vernier caliper, or micrometer, and it has a specific requirement for the size of test specimens [9]. In the strain gauge method, a perfect bond between the strain gauges and the sample is crucial. Additionally, the temperature and relative humidity have a significant impact on the test results [10]. To improve the precision of the measurement and to visualize the shrinkage of wood, some non-contact optical measurement methods such as digital image correlation and near-infrared spectroscopy have been applied for the determination of shrinkage strain [11,12,13,14]. However, the optical methods require real-time image acquisition for the wood drying process, which limits their application in wood industry.

Electrochemical methods are usually simple, rapid, accurate and sensitive characterization approaches, which have played an essential role in scientific research and industrial applications. The main principle is to accurately establish the relationship between the electrical parameters and the substance being measured, based on the electrochemical properties and changing rules of substances in the solution, and then conduct the qualitative and quantitative analyses for components. The electrochemical methods have been widely used in the fields of electrochemical power sources [15,16], chemical and biological sensors [17,18], corrosion of metals [19,20] and biotechnology [21]. However, the application of electrochemical methods in the field of wood science is scarce.

There is a vast amount of literature on the relationship between electrical properties and wood MC. The electrical properties of wood strongly depend on MC, exhibiting changes that span almost 10 orders of magnitude over the range of possible MCs [22]. As electrical resistance measurements provide information about wood MC, Brischke et al. designed a system for the long-term recording of wood MC with internal conductively glued electrodes [23]. Further, Brischke and Lampen determined resistance characteristics for a total of 27 wood-based materials and established a functional relation between electrical resistance and wood MC in a range between 15 and 50% MC [24]. In addition, the relationship between MC and material resistance is different in various MC ranges. It is affected by the wood species, experimental variables and calibration experiments [25,26,27]. Currently, industrial tests of commercial online MC meters have shown low accuracy of individual readings [28].

Taking into account the aspects mentioned above, some methods based on the electrical properties of wood have been used to determine wood MC. However, an electrochemical workstation is first used in the determination of wood MC, which has higher signal resolution and measuring stability. No previous research discusses the feasibility of electrochemical workstations in measuring shrinkage strain during wood drying. Thus, the presented study describes an electrochemical method to determine wood MC and shrinkage strain during drying, and the accuracy of this method is also evaluated.

## 2. Materials and Methods

### 2.1. Preparation of the Samples and Testing Equipment

Forty-five-year-old eucalyptus (*Eucalyptus exserta*) trees with a diameter of 40 cm at breast height were obtained from Tilestone town (110°42′ E, 22°09′ N), Gaozhou, China. Several flat-sawn lumbers with a dimension of 120 mm (tangential) × 25 mm (radial) × 900 mm (longitudinal) were sawn from one log, wrapped with a preservative film and stored in a freezer to keep them in the green condition. In this experiment, one flat-sawn lumber with an initial MC of 42% and no visible defects was chosen. Eight test specimens with the dimension of 120 mm (tangential) × 25 mm (radial) × 10 mm (longitudinal) were machined from the lumber and equally divided into two groups (Figure 1). One group was used to measure MC, and the other was for the determination of shrinkage strain.

A CHI760E electrochemical workstation test system (Shanghai Chenhua instrument equipment Co., LTD, Shanghai, China) was employed to measure the MC and shrinkage strain. The electrochemical workstation is an electronic instrument that controls the potential difference between the working electrodes and reference electrodes. In this study, one of the electrodes was connected to a working electrode, and the other was connected to a counter electrode and reference electrode together. The potentiostat controls the potential between the working electrode and the reference electrode and measures the current at the counter electrode so that a plot of potential vs. current can be created. The applied potential range was ±10 V, the current range was ±250 mA and the lower limit of the current measurement was below 50 pA. The electrochemical software of CHI760E (Shanghai Chenhua instrument equipment Co., LTD, Shanghai, China) was used for data acquisition, storage and processing. The diagram of the testing device and wood specimens is shown in Figure 1a, while the testing system can be observed in Figure 1b.

### 2.2. Determination of MC and Shrinkage Strain

The MC was measured by a current vs. time curve (i–t). The parameters were set as follows: the voltage was kept at a constant value of 1 V, the sampling interval was 0.1 s, the running time was 240 s and the sensitivity was set to 1 × 10^−6^ A/V. Before testing, both ends of the test specimens were sprayed with conductive coatings, which acted as contacts to enhance the conductivity between the wood and electrodes.

The shrinkage strain was measured by the current–time curve, combined with linear sweep voltammetry (LSV). The parameter settings of the current–time curve followed the measurement mentioned above regarding the moisture content. The parameter settings for LSV were as follows: the voltage ranged from −1 V to 1 V, the scan rate was 0.1 V/s, the sampling interval was 0.01 V and the sensitivity was 1 × 10^−6^ A/V. Before the electrochemical test, a conductive band with a thickness of 3 mm was sprayed on the surface of test specimens, as shown in Figure 1a.

Firstly, the initial weight for all test specimens and the initial length of the conductive band for shrinkage strain test specimens were obtained, and the electrochemical tests for all test specimens were performed. After that, the test specimens were dried at a constant temperature of 60 °C in a DKN611-type drying oven (Yamato Scientific Co., LTD, Tokyo Japan) to obtain different MC stages of wood specimens. The test specimens were taken out from the drying oven at drying times of 0.5 h, 1 h, 1.5 h, 2 h, 3 h and 4 h. The samples were weighed, followed by electrochemical measurements. For the shrinkage strain test specimens, the measurement of conductive band length was added. After finishing the tests, the specimens were oven dried to the absolute dry state and weighed. The moisture content, shrinkage strain and resistance change rate were calculated using Equations (1)–(3), respectively.
(1)$$\mathrm{MC}=\frac{M-M_{0}}{M_{0}}\times 100\%$$
where MC is the moisture content of test specimens, M is the weight of test specimens at different MCs and M_0_ is the weight after oven drying treatment of the test specimens.
(2)$$S=\frac{L_{0}-L_{i}}{L_{0}}\times 100\%$$
where S is the shrinkage strain of specimens, L_0_ is the initial length of the conductive band on the surface of test specimens and L_i_ is the length of the conductive band on the surface of test specimens at different moisture contents.
(3)$$\Delta R=\frac{R_{i}-R}{R_{i}}\times 100\%$$
where ΔR is the resistance change rate for shrinkage strain measurement specimens, R_i_ is the initial resistance at the MC of 42% and R is the resistance at different MCs.

## 3. Results and Discussion

### 3.1. The MC of Test Specimens

The MC of test specimens obtained with various drying times is shown in Figure 2. The MC of test specimens at the initial stage was about 42% and decreased with drying time. The changing trend almost remained the same for the specimens of MC and shrinkage strain. The mean value of MC at drying times of 0.5 h, 1 h, 1.5 h, 2 h, 3 h and 4 h was 31%, 23%, 17%, 12%, 8% and 5%, respectively. Therefore, all discussions and analyses after this section are based on these MC stages.

### 3.2. Analysis of MC Test Results

The electrical properties of wood can strongly depend on the MC, exhibiting changes that span almost ten orders of magnitude in the range of possible MCs [22]. The changes in current with time at different wood MC stages are shown in Figure 3a. As shown, the current had a positive correlation with moisture content. However, the phenomenon disappeared as the MC became lower than 12%. When MC decreased from 42% to 12%, the current was reduced from 1.2 × 10^−^^7^ A to 1.0 × 10^−8^ A, whereas the current showed nearly constant values at smaller MCs of 8% and 5%. This is because when the wood MC is lower than 10%, wood is similar to an insulator [22]. Thereby, the effect of MC on the electrical signal disappeared. The current slightly decreased with time and gradually reached a steady state at around 240 s. The resistance increased with the decrease in wood MC under a constant voltage of 1 V (Figure 3b). As the MC changed from 42% to 12%, the resistance increased from 1.0 × 10^7^ Ω to 1.2 × 10^8^ Ω, while the increasing trend continued at lower concentrations. For MCs of 8% and 5%, there was a tiny difference in resistance.

On the other hand, in the case where the MC decreased from 42% to 17%, the variation in electrical resistance with time was negligible, except for the first 20 s. For wood MC below 12%, the resistance fluctuated with time to a limited extent. This behavior may be related to the uneven distribution of the MC in wood specimens at the low-MC stage. According to the calculation formula of resistivity, the resistance was converted into resistivity, and the resistivity was about 10^6^~10^7^ Ω m at an MC range of 42% to 5%, which agrees with the results of the literature. As reported, the resistivity could be about 10^15^~10^16^ Ω m for oven-dried wood and 10^3^~10^4^ Ω m for wood at the fiber saturation point [22,29].

The relationship between MC and resistance can be observed in Figure 4. As it can be observed, the electrical resistance increased in a nonlinear way with the decrease in MC, and there was an apparent increasing trend as MC was reduced from 31% to 17%. Similar results reported by Barański et al. showed a nonlinear dependence of wood resistance on the moisture content [25]. In this study, a polynomial fitting curve was employed, and the determination coefficient was 0.937, which indicated that the resistance of wood specimens was capable of explaining more than 93.7% of the wood MC. The relationship between MC and resistance, according to the fitting curve, can be expressed as Equation (4). Thus, wood MC can be obtained directly by the changes in resistance using the electrochemical approach.
(4)$$\mathrm{MC}=47.59-1.17R+0.013R^{2}-4.8\times 10^{-5}R^{3}$$
where MC is the moisture content of wood test specimens, and R is 10^−6^ times the electrical resistance of wood test specimens.

### 3.3. Analysis of Shrinkage Strain Test Results

The variation in the current and resistance at different MC stages for measuring shrinkage strain is shown in Figure 5. The current increased significantly with the decrease in moisture content, increasing from about 0.014 A to 0.028 A as the MC dropped from 42% to 5%. In contrast, the resistance decreased gradually with decreasing MC. Furthermore, the variation in electrical resistance ranged from 70 Ω to 35 Ω within the measured range of the moisture content. However, this phenomenon conflicts with the inverse relationship between electrical resistance and MC observed in Figure 3b. The reason can be explained as follows: comparing Figure 3b with Figure 5b, the electrical resistance in test specimens of MC was about 6~7 orders of magnitude higher than in the test specimens of shrinkage strain. This observation revealed that the conductivity of the conductive silver paint used in measuring shrinkage strain was much better than that of the wood itself, and the electrical current could move through the conductive band with low resistivity instead of wood tissue. Therefore, the resistance change with MC in measuring shrinkage strain was determined by the conductive band. Generally, the value of resistance decreased with the decreasing length of the electrical conductor. With the shrinkage of the wood, the length of the conductive band shortened, and in conjunction with the accumulation of the conductive silver paint, the electrical resistance gradually decreased with the wood shrinkage generated by the decrease in MC.

The LSV method was used to explore the changes in the current and resistance at different voltages. The current and resistance changes with the MC at a range of −1~1 V are presented in Figure 6. As observed in Figure 6a, the slope of the lines represents the inverse of the resistance, which varies with the changes in MC. The slope also confirms the variation in resistance at different MC stages. As seen in Figure 6b, the resistance remained constant when the voltage changed from −1 V to 1 V at each MC, but there was a great difference at different MCs. A particular situation can be seen at the MCs of 8% and 5%, where the curves of resistance vs. voltage overlap, indicating that the change in electrical resistance generated by shrinkage was very small at this MC stage. Moreover, Figure 5 also provides a piece of evidence as the voltage does not affect the testing results of electrical resistance. Thus, the activation voltage was free to choose at the coverage of −1 V to 1 V in this study.

Figure 7 shows the variation in the resistance change rate and shrinkage strain at different MCs. Wood shrinkage was strongly connected with the water in the wood; once the wood MC dropped to the fiber saturation points, shrinkage occurred. It is widely accepted that the fiber saturation point is at an MC of approximately 30% for most wood species, which is a turning point for wood physics and mechanical properties [30,31]. Therefore, the shrinkage strain in this study was discussed from the MC of 31%, and no shrinkage was considered at the initial MC of 42%. As observed in Figure 7, both the shrinkage strain and resistance change rate increased with the decrease in wood MC, especially for the MC range of 23% to 8%, where the shrinkage strain and resistance change rate increased by 4% and 30%, respectively. As observed, the shrinkage of wood clearly increased with the decrease in MC below the FSP [9,32]. From Section 3.3, the electrical resistance gradually decreased with the wood shrinkage generated by the decrease in MC, and thus the resistance change rate increased with decreasing MC. In the case of shrinkage strain, its value increased from 1% at an MC of 23% to 5% at an MC of 8%, and the matching values for the resistance change rate changed from 17.5% to 47.5%. These results demonstrate a close correspondence between shrinkage strain and the resistance change rate.

In order to describe the correlations between shrinkage strain and the resistance change rate, the fitting curve between the two parameters is presented in Figure 8. As shown, the shrinkage strain increased exponentially with the increasing resistance change rate. The regression equation between shrinkage strain and the resistance change rate was described by Equation (5), and the determination coefficient reached 0.985, indicating that the prediction success rate of shrinkage strain was as high as 98.5% using this equation. This result provides a decent approach for the determination of wood shrinkage strain and also shows the feasibility of the electrochemical method in determining the wood shrinkage behavior.
(5)$$S=0.852+0.021e^{0.113\Delta R}$$
where S is the shrinkage strain, and ΔR is the resistance change rate.

## 4. Conclusions

The present research discussed the applicability of electrochemical methods for the determination of MC and shrinkage strain in wood drying, evaluating the precision as well.

According to the test results for MC, the electrical resistance clearly increased with the decreasing wood MC, especially in the MC range of 42% to 12%. A polynomial fitting curve with a determination coefficient of 0.937 was employed to describe the relationship between MC and electrical resistance.

In the case of the measurement of shrinkage strain, the electrical resistance gradually decreased with the decrease in MC, and the voltage did not affect the results of electrical resistance. The resistance change rate was further chosen as the correlation parameter to characterize shrinkage strain. The shrinkage strain increased exponentially with the increase in the resistance change rate. An exponential regression equation with the determination coefficient of 0.985 was determined to describe the correlation between shrinkage strain and the resistance change rate.

The findings of this article demonstrate the feasibility of the electrochemical approach to determine the MC and shrinkage strain in the wood drying process. Additional studies will be conducted on full-size specimens to achieve the applicability of this method in industrial production.

## Figures and Tables

**Figure 1 polymers-14-00778-f001:**
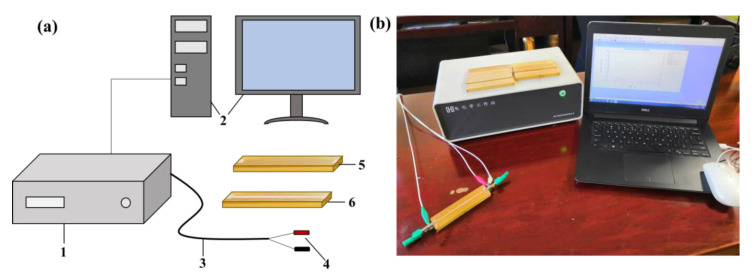
Diagram of the testing device and wood specimens (**a**): 1—electrochemical workstation; 2—computer host and monitor; 3—electric wire; 4—conductive clip; 5—test specimen for MC; 6—test specimen for shrinkage strain. Testing system of the electrochemical method (**b**).

**Figure 2 polymers-14-00778-f002:**
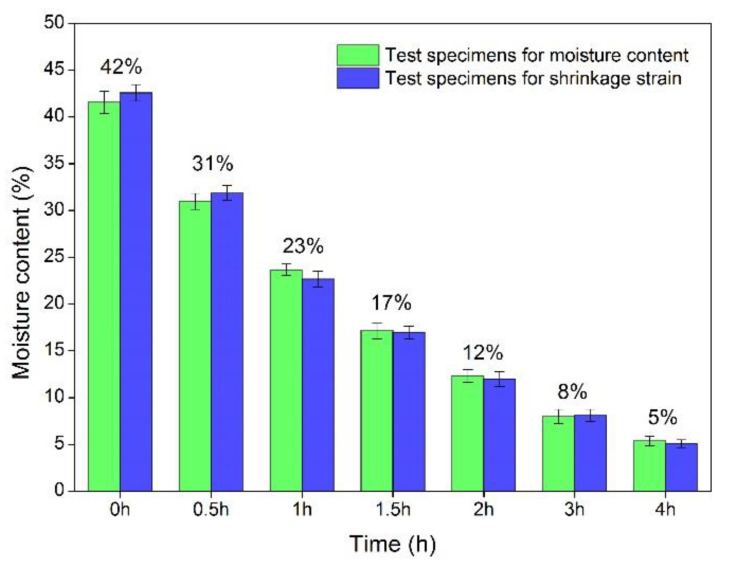
Changes in MC with the applied drying times.

**Figure 3 polymers-14-00778-f003:**
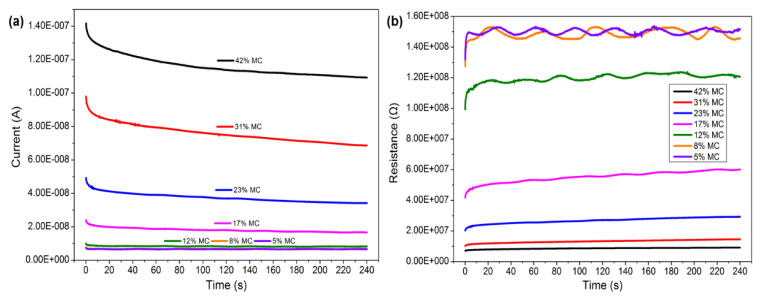
Changes in current (**a**) and resistance (**b**) over time at a potential of 1 V for the measurement of moisture content.

**Figure 4 polymers-14-00778-f004:**
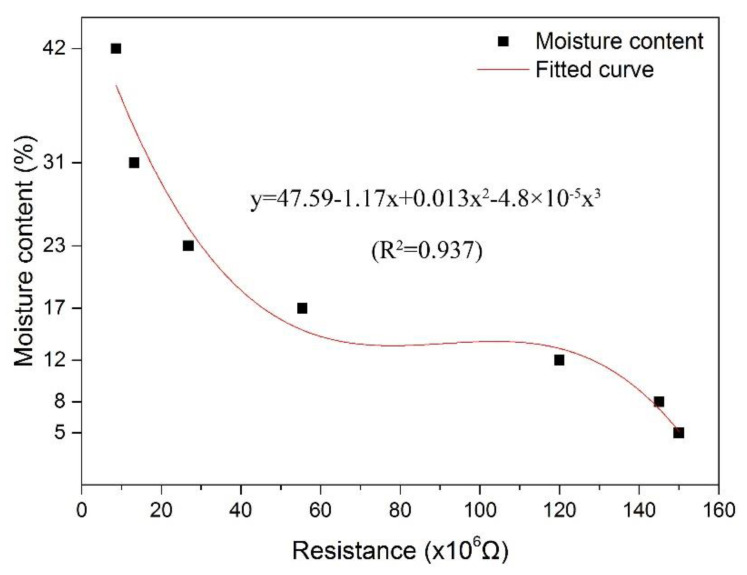
The fitting curve for wood MC to electrical resistance.

**Figure 5 polymers-14-00778-f005:**
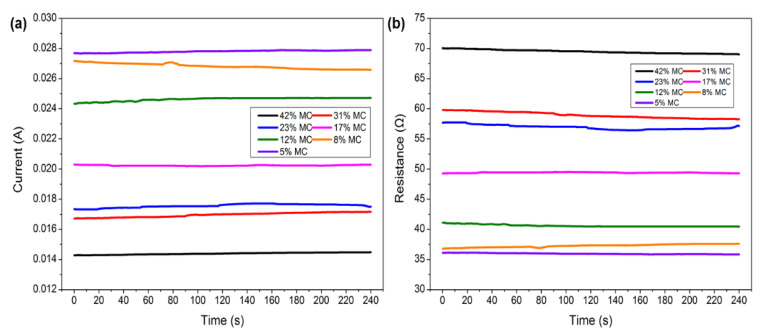
Current (**a**) and resistance (**b**) change over time under a constant voltage of 1 V for measuring shrinkage strain.

**Figure 6 polymers-14-00778-f006:**
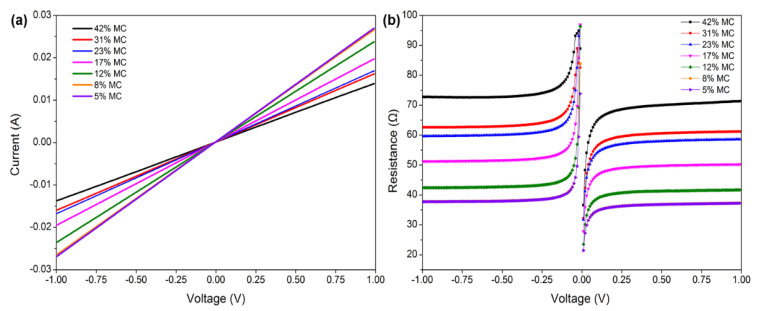
Plots of current (**a**) and resistance (**b**) with voltage for measuring shrinkage strain.

**Figure 7 polymers-14-00778-f007:**
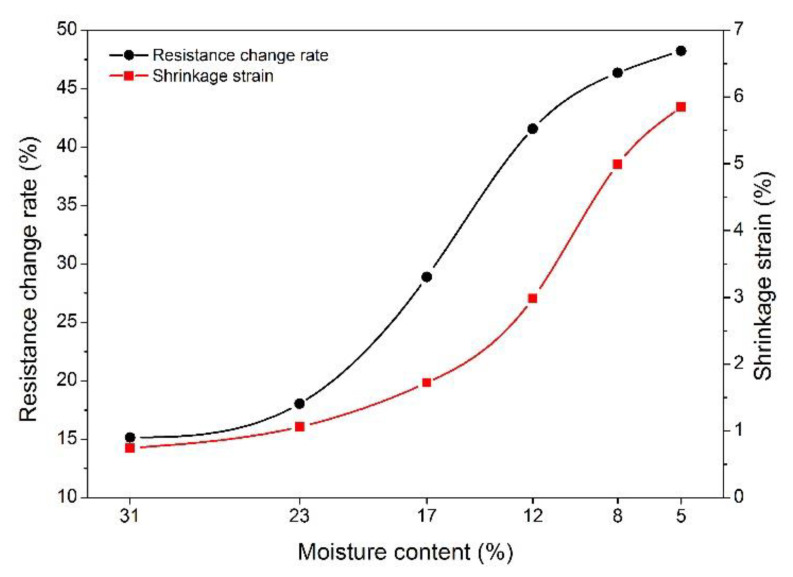
Resistance change rate and shrinkage strain at different MC stages.

**Figure 8 polymers-14-00778-f008:**
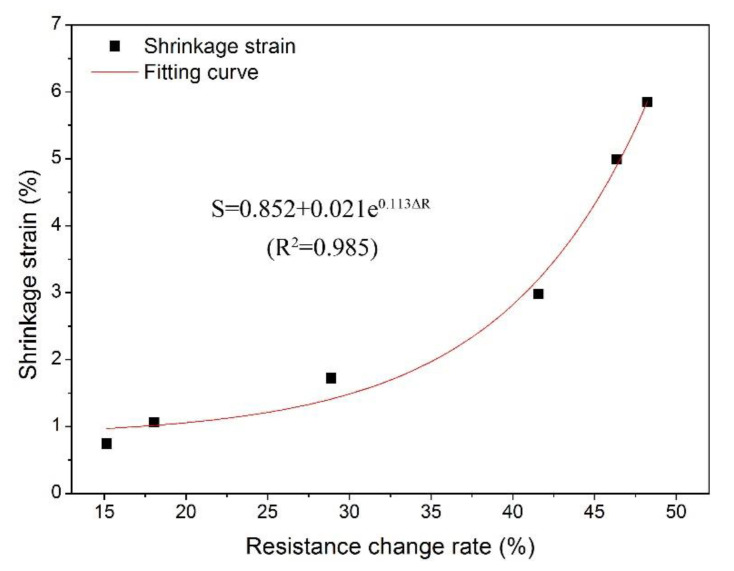
Fitting curve for the resistance change rate to shrinkage strain of wood.

## Data Availability

The data presented in this study are available upon request from the corresponding author.

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
