# Peer review of "Determination of Moisture Content and Shrinkage Strain during Wood Water Loss with Electrochemical Method"

_polymers, 2022, doi:10.3390/polym14040778_

Round 1
Reviewer 1 Report
Thank you for submitting your paper. The work done here draws attention to a significant subject in Moisture content and shrinkage strain in wood. I have found the paper to be interesting. However, several issues need to be addressed properly before the paper is being considered for publication. My comments including major and minor concerns are given below:
- Please consider reviewing the abstract and highlight the novelty, major findings, and conclusions. I suggest reorganizing the abstract, highlighting the novelties introduced. The abstract should contain answers to the following questions:
- What problem was studied and why is it important?
- What methods were used?
- What conclusions can be drawn from the results? (Please provide specific results and not generic ones).
- The abstract must be improved. Please use numbers or % terms to clearly shows us the results in your experimental work. Please expand the abstract.
- Please consider reporting on studies related to your work from mdpi journals.
- The introduction must be expanded, please consider improving the introduction, provide more in-depth critical review about past studies similar to your work, mention what they did and what were their main findings then highlight how does your current study brings new difference to the field.
- Line 149-150 why? Please explain this trend further and support with references.
- Equation 4 format is not the same as previous ones, please make sure to unify the format of all formulas.
- The quality of figures 3 and 4 can be improved.
- Line 182 “which increased double as moisture content dropped..” requires rephrasing, the sentence does not read well.
- Lines 180-197 there is a lot of statements, decreased, increased, …etc but not followed by scientific justification and explanation of the trends. This must be fixed and updated. Explain each of the observations and support with references.
- The results are merely described and is limited to comparing the experimental observation and describing results. The authors are encouraged to include a more detailed results and discussion section and critically discuss the observations from this investigation with existing literature.
- Conclusion can be expanded or perhaps consider using bullet points (1-2 bullet points) from each of the subsections.
Reviewer 2 Report
The subject is interesting but made on small samples (so what is the interest in relation to reproducibility in industry), it also lacks a more interesting photo of a sample.
1. Why are the samples stored in a freezer, is it obtained by a standard?Because it will not modify the pores and the humidity?
2. Why is the oven set at 60°C? standard?
3. How is the starting humidity 42% determined?
4. There is an error on lines 134-135, the value 42% should be removed.
5. There is a lack of references on the choices of the experimentation and the measurement parameters. Why -1V to 1V? Why 240s?
6. We see that some developments in Figure 3, 5 continue to progress, Why? How is resistance determined? Is the trial time correct?
7. Finally, the correlations seem interesting but are they reproducible on larger samples?
Many results described but little analyzed.
Reviewer 3 Report
The manuscript describes the correlation between wood moisture content during drying of small samples with the electrical properties and shrinkage of the material. While the samples are too small for the drying process to simulate realistic industrial drying of wood, I find it incomprehensible that no references are given to the vast literature on the relationship between electrical properties and wood moisture content. This makes the study unsuitable for publication in its current version. The authors should carefully read through the literature and describe the novelty of this study (or whether it is a replication of previous research) before the manuscript will have any value to the reader.
Below is a list of various literature references detailing investigations of electrical properties in wood and their dependence on wood moisture content:
Brischke, C. and Lampen, S.C. (2014). Resistance based moisture content measurements on native, modified and preservative treated wood. Eur. J. Wood Prod. 72: 289–292 https://doi.org/10.1007/s00107-013-0775-3.
Brischke, C., Rapp, A.O., and Bayerbach, R. (2008). Measurement system for long-term recording of wood moisture content with internal conductively glued electrodes. Build. Environ. 43: 1566–1574 https://doi.org/10.1016/j.buildenv.2007.10.002.
Davidson, R. (1958). The effect of temperature on the electrical resistance of wood. For. Prod. J. 8: 160–164.
Du, Q.P., Geissen, A., and Noack, D. (1991). Widerstandskennlinien einiger Handelshölzer und ihre Meßbarkeit bei der elektrischen Holzfeuchtemessung (Moisture/resistance characteristics of some commercial wood species and their measurability by DC type moisture meters). Holz Roh. Werkst. 49: 305–311 https://doi.org/10.1007/BF02663796.
Hiruma, J. (1915). Experiment of the electric resistance in wood. In: Extracts from the bulletin of the forest experiment station, Meguro, Tokyo. Bureau of Forestry, Department of Agriculture and Commerce, Tokyo, Japan, pp. 59–65.
Hasselblatt, M. (1926). Der Wasserdampfdruck und die elektrische Leitfähigkeit des Holzes in Abhängigkeit von seinem Wassergehalt (Water vapour pressure and electrical conductivity of wood in dependence of water content). Z. Anorg. Allg. Chem. 154: 375–385 https://doi.org/10.1002/zaac.19261540133.
James, W.L. (1961). Calibration of electric moisture meters for Jack and Red pine, Black spruce, Paper birch, Blask ash, Eastern hemlock, and Bigtooth aspen. USDA Forest Service, Forest Products Laboratory, Madison, WI, USA, pp. 1–7.
James, W.L. (1988). Electric moisture meters for wood. Forest Products Laboratory General Technical Report FPL-GTR-6. US Forest Service, Forest Products Laboratory, Madison, WI, USA, pp. 1–17.10.2737/FPL-GTR-6
Keylwerth, R. and Noack, D. (1956). Über den Einfluß höherer Temperaturen auf die elektrische Holzfeuchtigkeits-messung nach dem Widerstandsprinzip (On the influence of elevated temperatures on the electrical wood moisture content determination by the principle of reistance). Holz Roh Werkst 14: 162–172 https://doi.org/10.1007/BF02617621.
Lin, R.T. (1965). A study of electrical conduction in wood, Ph.D. Dissertation. State College of Forestry at Syracuse University. Syracuse., New York.
Myer, J.E. and Rees, L.W. (1926). Electrical resistance of wood with special reference to the fiber saturation point, Vol. 26. Syracuse University, Syracuse, NY, USA, pp. 1–22.
Nusser, E. (1938). Die Bestimmung der Holzfeuchtigkeit durch Messung des elektrischen Widerstandes. Holz Roh Werkst 1: 417–420 https://doi.org/10.1007/BF02605258.
Sharma, S.K., Shukla, S.R., and Kamala, B.S. (1997). Studies on DC electrical resistivity of plantation grown timbers. Holz Roh Werkst 55: 391–394 https://doi.org/10.1007/s001070050252
Stamm, A.J. (1927). The electrical resistance of wood as a measure of its moisture content. Ind. Eng. Chem. 19: 1021–1025 https://doi.org/10.1021/ie50213a022.
Stamm, A.J. (1929). The fiber-saturation point of wood as obtained from electrical conductivity measurements. Ind. Eng. Chem. Anal. Ed. 1: 94–97 https://doi.org/10.1021/ac50066a021.
Takechi, O. and Inose, O. (1953). Analysis on the fundamental properties of electric resistance of wood II—on the electric resistance of wood in relation to the moisture content and the temperature. Sci. Rep. Matsuyama Agric. College. 10: 13–35.
Vermaas, H. (1982). DC resistance moisture meters for wood: Part I: review of some fundamental considerations. S Afr. For. J. 121: 88–92 https://doi.org/10.1080/00382167.1982.9628815.
Reviewer 4 Report
Manuscript: polymers-1577982-peer-review-v1
Title: „ Determination of Moisture Content and Shrinkage Strain during Wood Water Loss with Electrochemical Method”
The manuscript contains 9 pages of text with tables, figures and a reference list.
The submitted text is in line with the subject of the Polymers magazine.
After reading the text, in general, the scientific quality of the publication is good, although it is not a text of great scientific importance. I believe that the text requires minor corrections that do not significantly affect the quality of the entire publication, but make the text easier to read and understand.
There are many devices on the market for quick wood moisture determination based on resistance measurement. In the submitted manuscript, I lacked information that the proposed method and algorithms can be implemented in practice in existing or new measuring devices. I suggest you add it in the final summary. The practical use of research and results should be emphasized.
I do not judge the correctness of the language as English is not my mother tongue.
General comments.
Title. Overall, it is correct and corresponds to the content.
Abstract. It is spelled correctly. It contains the most important information about the research and the results obtained.
Introductions. The literature review is short, but it contains the most important information related to the research topic. However, I am missing one main statement indicating deficiencies in the literature or technologies, which will be an indication for the research undertaken.
Materials and Methods.In my opinion, the methodology is well written and allows the research to be repeated by both authors and other researchers.
Line 105-107. The tape is not visible in the picture. Improve drawing quality or insert a photo of the sample with the tape attached.
Results and Discussions.The results obtained are well described and discussed.In my opinion, however, the discussion is too weak.I suggest supplementing it with more comparisons to other tests and measurement methods - incl.indicate whether your method is more accurate than others.
Figure 3-8. In all drawings, the descriptions and markings are made in a very small font and are illegible. Improve quality - larger font.
Figure 4 and 8. Transfer the tables from the graphs to the content. They are illegible
Conclusions. They are generally correct and are based on the results. However, I propose to supplement them with information on the introduction of the method into industry, designing a measuring device, etc.
References. The authors cite 25 items in the world literature from the last 20 years - the oldest from 2005. These publications are related to the subject of the manuscript.
Round 2
Reviewer 1 Report
The authors provided the answers to the comments from the first round of review and made sufficient changes in the manuscript according to these comments. I recommend this manuscript for a publication in its present form.
Author Response
The authors provided the answers to the comments from the first round of review and made sufficient changes in the manuscript according to these comments. I recommend this manuscript for a publication in its present form.
Response: Thanks so much for your comments.
Reviewer 2 Report
The corrections made are fine with me.Author Response
The corrections made are fine with me.
Response: Thanks so much for your comment.
Reviewer 3 Report
The manuscript has been improved, but there are still important information missing. First of all, the fundamental measurement principle is not very well described. The device name, brand and settings are described, but how it functions is not clear at all.
Secondly, the wood specimens are all very small compared with full-scale lumber. Therefore, it is not clear if the measurement device (or principle) can be used on industrial scale - even though the authors in the introduction uses a lot of time on how other methods are either expensive or cumbersome to use. For instance, the electrical resistance method is described on page 2: "The common electrical resistance method in wood MC determination is required that two electrodes inserted in the wood." This is not true, since contact electrodes can be used - and on the small specimens used in this study, the electrical resistance method (using contact electrodes) would work fine in the moisture range investigated in this study (both methods fail below 12 %MC unless very high electrical resistances can be determined).
The authors have thus failed to appropriately address my original comment: "The authors should carefully read through the literature and describe the novelty of this study (or whether it is a replication of previous research) before the manuscript will have any value to the reader." The novelty is not described - i.e. how the method employed in this study functions, is novel and provides better results than other methods.
Author Response
The manuscript has been improved, but there are still important information missing. First of all, the fundamental measurement principle is not very well described. The device name, brand and settings are described, but how it functions is not clear at all.
Response: Thanks for your comments. We have added the fundamental measurement principle of electrochemical workstation. The details can be seen in the revised manuscript Line 106-112 highlighted with green color.
Secondly, the wood specimens are all very small compared with full-scale lumber. Therefore, it is not clear if the measurement device (or principle) can be used on industrial scale - even though the authors in the introduction uses a lot of time on how other methods are either expensive or cumbersome to use. For instance, the electrical resistance method is described on page 2: "The common electrical resistance method in wood MC determination is required that two electrodes inserted in the wood." This is not true, since contact electrodes can be used - and on the small specimens used in this study, the electrical resistance method (using contact electrodes) would work fine in the moisture range investigated in this study (both methods fail below 12 %MC unless very high electrical resistances can be determined).
Response: Thanks for your comments. We have deleted the inappropriate sentence. As for the question of if the measurement device (or principle) can be used on industrial scale, the results from this study has been proved the measurement principle is absolutely feasible. But there is still a gap between small wood specimens and full-scale wood. Thus, if this method used is wood industry, it still needs to do more complementary work. We also have mentioned that in the part of Conclusions. The details can be seen in the revised manuscript Line 283-286. “The findings of this article demonstrated the feasibility of the electrochemical approach to determine the MC and shrinkage strain in wood drying process. Additional studies will be conducted on full-size specimens to achieve the applicability of this method in industrial production.”
The authors have thus failed to appropriately address my original comment: "The authors should carefully read through the literature and describe the novelty of this study (or whether it is a replication of previous research) before the manuscript will have any value to the reader." The novelty is not described - i.e. how the method employed in this study functions, is novel and provides better results than other methods.
Response: Thanks. Based on your comments, we have added some description about the novelty of this study. The details can be seen in the revised manuscript Line 85-89 highlighted with green color.
Round 3
Reviewer 3 Report
The manuscript has been appropriately revised to warrant publication.